# The Impact of Novel Anticoagulants on the Upper Gastrointestinal Tract Mucosa

**DOI:** 10.3390/medicina56070363

**Published:** 2020-07-21

**Authors:** Lubomir Mihalkanin, Branislav Stancak

**Affiliations:** 1Gastroenterology Department, Gastro-LM, 080 01 Presov, Slovakia; mihalkanin@gmail.com; 2Cardiology Department, East Slovakia Institute for Cardiovascular Diseases, 040 11 Kosice, Slovakia

**Keywords:** novel anticoagulants, gastrointestinal bleeding, *Helicobacter pylori*, proton pump inhibitors, endoscopic biopsy

## Abstract

*Background and objectives:* Although treatment with novel oral non-vitamin K antagonist 3anticoagulants (NOACs) is associated with an overall decrease in hemorrhagic complications compared to warfarin, the incidence of gastrointestinal bleeding remains contradictory. *Materials and Methods:* After the exclusion of patients with pre-existing pathological lesions in the upper gastrointestinal tract (GIT) on esophageal-gastroduodenoscopy (EGD) at entry, a cohort of 80 patients (mean age of 74.8 ± 2.0 years) was randomly divided into four equivalent groups, treated with dabigatran, rivaroxaban, apixaban, or warfarin. Patients were prospectively followed up for three months of treatment, with a focus on anamnestic and endoscopic signs of bleeding. In addition, bleeding risk factors were evaluated. *Results:* In none of the patients treated with warfarin or NOACs was any serious or clinically significant bleeding recorded within the follow-up period. The incidence of clinical bleeding and endoscopically detected bleeding in the upper GT after three months of treatment was not statistically different among groups (χ^2^ = 2.8458; *p* = 0.41608). The presence of *Helicobacter pylori* (HP) was a risk factor for upper GIT bleeding (*p* < 0.05), while the use of proton pump inhibitors (PPIs) was a protective factor (*p* = 0.206; Spearman’s correlation coefficient = 0.205). We did not record any post-biopsy continued bleeding. *Conclusions:* No significant GIT bleeding was found in any of the treatment groups, so we consider it beneficial to perform routine EGD before the initiation of any anticoagulant therapy in patients with an increased risk of upper GIT bleeding. Detection and eradication of HP as well as preventive PPI treatment may mitigate the occurrence of endoscopic bleeding. Endoscopic biopsy during the NOAC treatment is safe.

## 1. Introduction

Antithrombotics and anticoagulants are some of the most frequently used drugs worldwide. Paradoxically, they also rank among the top medical drugs of all categories for serious adverse events, especially bleeding. Novel anticoagulants (NOACs) of the second generation directly inhibiting factor Xa (rixaroxaban, apixaban) or thrombin (dabigatran) have recently been introduced into broad clinical practice. They can be administered at fixed doses without the need for laboratory monitoring. They have reached a strong position in anticoagulant treatment, whether in venous thrombosis and pulmonary embolism or in the prevention of systemic embolism, including stroke, in non-valvular atrial fibrillation.

Although the incidence of gastrointestinal tract (GIT) hemorrhage has slightly decreased in recent years (upper GIT 100/100,000 and lower GIT 20/100,000), mortality remains constant. Relapses are very common and occur in 20% of patients [1]. Bleeding into the GIT seems to be an “Achilles’ heel” in treatment with NOAC. Bleeding occurs in approximately 1.5–2% of patients treated with NOACs per year. One third of these bleedings are located in the GIT, more seriously in the upper GIT than at other sites [2,3,4,5]. A recently reported 90-day mortality in patients with serious GIT bleeding was up to 16.2% [6]. The risk factors for bleeding are considered previous bleeding in the GIT, history of gastroduodenal ulcer, age, hypertension, heart failure, smoking, male gender, anemia, chronic obstructive bronchopulmonary disease, impaired renal and liver function, and concomitant use of antiplatelets agents or non-steroidal anti-inflammatory drugs. The risk of GIT bleeding is the highest within the first three months of treatment [2,3,7,8].

Holster (2013) [9], in his systematic review of 43 studies including 154,578 patients, revealed that administration of NOAC was associated with a mild but significant increase in GIT bleeding compared to standard treatment or placebo (RR 1.45). During the follow-up period, ranging from 3 weeks to 31 months, the GIT bleeding rate was 1.5%, of which 89% were major bleedings. These were defined as a decrease in Hb ≥ 20 g/L for 24 h, required administration of ≥2 red blood cell units, or surgical intervention and fatal bleeding [6].

Due to great heterogeneity in clinical trials, the determination of which NOAC is the safest is difficult. However, by indirect comparative analysis of studies, dabigatran at 2 × 150 mg and rivaroxaban administered once daily seem to increase the risk of GIT bleeding 1.5-fold compared to warfarin, while a similar incidence of GIT bleeding as that with warfarin has been observed with dabigatran 2 × 110 mg and apixaban [2,10,11,12,13,14].

Current published studies have several limitations. All of them are retrospective or are meta-analyses. Another limitation is that, according to the inclusion criteria, studies enroll only patients with a low risk of GIT bleeding, although in routine clinical practice, high-risk patients occur at a rate of 25–40% and therefore the risk of bleeding is expected to be 3- to 15-fold higher [10]. Most studies only report partial endpoints and bleeding rates without further specification. Moreover, patients with a recent history of peptic ulcer disease or patients with increased risk of GIT bleeding due to thrombocytopenia or coagulopathy have been excluded from many of these studies [15,16].

Another limitation is that the majority of studies report only clinically significant bleeding, which is associated with a decrease in Hb of more than 20 g/L or require substitution of losses with two or more units of red blood cells [15].

Less significant bleeding that does not meet the criteria of major bleeding, but requires clinical intervention (like an unplanned visit to a physician, necessary change in treatment, or patient discomfort), as well as minor bleeding (not meeting the criteria of major and clinically significant non-major bleeding) is not reported [15,17]. All these factors finally have led to the underrating of clinically relevant bleeding. Moreover, none of these studies provide a prediction of the bleeding risk, as well as an estimation of the true effect of warfarin or NOAC on normal or already affected gastrointestinal mucosa. 

The aim of the study was to investigate the impact of three NOACs (rivaroxaban, dabigatran, and apixaban) on the upper gastrointestinal mucosa and compare their impact with warfarin. A secondary objective was to identify the risk bleeding factors in patients prior to initiating oral anticoagulant treatment.

## 2. Materials and Methods

In the period between July 2014 and January 2018, we examined 156 ambulatory patients with atrial fibrillation or deep vein thrombosis indicated for anticoagulant treatment. If patients agreed with the study protocol, they underwent esophageal-gastroduodenoscopy (EGD) prior to the initiation of oral anticoagulant therapy. All examinations were performed by a single experienced examiner (LM) using an Olympus Exera II CV 180 HDTV videogastrocope. Patients with pre-existing lesions, e.g., the presence of erosive mucosal changes (erosions and ulcerative lesions) or mucosal signs of bleeding at baseline endoscopic examination, were excluded from further study. It was assumed that these patients would be more prone to bleeding after the initiation of anticoagulant treatment. A total of 100 patients were enrolled, i.e., first series of 20 patients with negative EGD in each NOAC group (dabigatran, rivaroxaban, apixaban) and 40 patients in the warfarin group. Patients were assigned to groups randomly. The dosage of warfarin was based on the regular monthly measurements of the International Normalized Ration (INR), while the dosage of NOACs followed the Summary of Product Characteristics for any given drug. In elderly patients over the age of 75 years, the dosage of dabigatran was reduced from 150 to 110 mg twice daily, while in rivaroxaban, the standard dose of 20 mg was reduced to 15 mg once daily. The standard dose of apixaban was 5 mg twice daily; in elderly patients, this was reduced to 2.5 mg twice daily. None of the patients had impaired renal function requiring an adjustment of dosage. The EGD examination was repeated three months after the initiation of therapy. During endoscopy, we performed photographic documentation of five different anatomical sites: Distal esophagus, stomach body, stomach antrum, duodenal bulbus, and the D2 duodenum (Figure 1). 

In the initial examination, we evaluated anamnestic data (summarized in Table 1). From these data, we calculated the CHA_2_DS_2_-VASc score (C—cardiac failure, i.e., documented moderate to severe systolic dysfunction, H—hypertension, A—age ≥ 75 years, D—diabetes mellitus, S—stroke, V—vascular disease, A—age (age 65–74), Sc—sex category), and HAS-BLED score (H—hypertension, A—kidney/liver abnormal function, S—stroke, B—bleeding history or predisposition, L—INR lability, E—elderly, D—drug).

During the subsequent EGD examinations, performed after three months of anticoagulant treatment, we looked for the presence of bleeding and/or any mucosal changes. To describe the lesion, we used the terminology of the European Society of Gastrointestinal Endoscopy (ESGE) [18]. The findings were evaluated semi-quantitatively. We performed photographic documentation similarly as in the initial examination. We evaluated a need for the additional use of proton pump inhibitors (PPIs) as well as other alterations to medical therapy. The presence of *Helicobacter pylori* (HP) (if not documented) was tested with a rapid urease test from gastric tissue samples (antrum and body) taken during the endoscopy. In patients who were treated with warfarin, a blood sample for INR was taken on the day of the control EGD to determine the effectiveness of this anticoagulant therapy. The first 20 patients with a therapeutic value of the INR (2.0–3.0) were taken into the evaluation, so the total number of patients in the study was 80 and all groups were equivalent.

Statistical processing of the results consisted of a one-dimensional analysis of variance, followed by post-hoc Scheffe’s and Newman–Keul’s tests for group average comparison. In the absence of distribution normality, a non-parametric Mann–Whitney’s U test was used. The degree of tightness of statistical dependence was determined by the Spearman’s correlation coefficient (R); the statistical significance of the correlation coefficient was examined by Fisher’s z-transformation. The resulting data are presented as the arithmetic mean ± standard deviation. Alternative data were evaluated by the chi-square test (χ^2^); in the case of low numbers in the cell, we used Yate’s correction. *p* values less than 0.05 was considered as a level of significance in all the above-mentioned statistical methods.

**Ethical statement**: All patients gave their informed consent for inclusion before they participated in the study. The study was conducted in accordance with the Declaration of Helsinki, and the protocol was approved by the Ethics Committee of PJ Safarik University in Kosice, Slovakia (ref. No. 4021/2014-Vv, Approval Date: 3 June 2014).

## 3. Results

When comparing the input data of patient groups (Table 1) using one-dimensional variance analysis, no difference was found between the NYHA class among groups as well as in the CHA_2_DS_2_-VASc score (*p* = 0.345) (Figure 2). In the HAS-BLED score (Figure 3), we found a significant difference between groups (*p* = 0.038). The highest HAS-BLED score was found in patients treated with dabigatran and the lowest score in those treated with warfarin (*p* = 0.005). There was no difference in the occurrence of HP among groups (χ^2^ = 0.5011, *p* = 0.918).

When comparing the incidence of bleeding in the upper GIT only, the differences were statistically insignificant for all drugs used (χ^2^ = 2.8458, *p* = 0.41608) (Table 2, Figure 4).

When analyzing PPI treatment in individual groups, we found that the treatment of PPI was more frequently used in patients on rivaroxaban compared with other groups (χ^2^ = 12.9167, *p* = 0.00482). In this group, more patients were treated for gastroesophageal reflux disease (GERD) or had a history of ulcer disease. The correlation between PPI use and the incidence of bleeding was insignificant (*p* = 0.206, *R* = −0.205). However, a borderline value of the Spearman’s correlation coefficient may indicate a trend for a decrease in bleeding episodes in patients receiving PPI.

According to regression analysis, the incidence of bleeding was not dependent on the age of the patients or on the dose of NOAC. Bleeding was found more frequently in patients with positive HP overall or receiving warfarin or dabigatran (*p* < 0.05). We also found that the CHA_2_DS_2_-VASc score correlated well with the HAS-BLED score (*p* < 0.05, *R* = 0.400). We were not able to confirm a higher incidence of bleeding in patients with diabetes mellitus.

## 4. Discussion

NOACs have become a common and effective cornerstone in the prevention and treatment of venous thromboembolism and the prevention of stroke in patients with atrial fibrillation. They have shown better efficacy and safety compared to warfarin in a reduced incidence of systemic embolism or stroke at 19% (RR 0.81, 95% CI 0.73–0.91, *p* < 0.0001), especially hemorrhagic stroke at 51% (RR 0.49, 95% CI 0.38–0.64, *p* < 0.0001). They also reduce patient mortality at 10% (RR 0.90, 95% CI 0.85–0.95) [12].

However, bleeding remains an important adverse effect of anticoagulant therapy. The incidence of bleeding complications has decreased overall, but a meta-analysis of four early studies including 42,411 patients showed a more frequent incidence of GIT bleeding with NOAC treatment compared with warfarin (Ruff, 2013) [19]. Similar results were shown in a meta-analysis of 43 studies (154,578 patients, RR 1.45) (Holster, 2013) [9]. On the contrary, later studies show a higher incidence of bleeding with warfarin than NOAC (Cangemi, 2017 [20], Xue, 2019 [21]). Another meta-analysis of 28 randomized trials (Miller, 2017) [15] including 129,357 patients did not reveal any difference in the incidence of GIT bleeding between NOACs overall and conventional anticoagulant therapy: Major bleeding (1.5% vs. 1.3%, respectively, RR 0.98, 95% CI, 0.80–1.21) and upper GIT bleeding (1.5% vs. 1.6%, respectively, RR 0.96, 95% CI, 0.77–1.20). On the contrary, when comparing NOACs individually with warfarin, an increased incidence of major GIT bleeding was found in dabigatran (2.0 vs. 1.4%, RR 1.27, 95% CI, 1.04–1.55) and rivaroxaban (1.7% vs. 1.3%, RR 1.4, 95% CI, 1.15–1.70) groups. No difference was found with apixaban versus warfarin treatment (0.6% vs. 0.7%, RR 0.81, CI 95%, 0.64–1.02). The same conclusion was drawn from a large retrospective population study conducted by the Mayo Clinic [22] comparing different NOACs’ impact on the incidence of major GIT bleeding. When comparing the incidence of bleeding in patients treated with apixaban vs. dabigatran (31,574 patients) and apixaban vs. rivaroxaban (13,130 patients), a lower incidence of bleeding was found with apixaban (RR 0.39 and 0.33, CI 95%, respectively). When comparing rivaroxaban and dabigatran, an increase of 20% of GIT bleeding events was reported with rivaroxaban (RR 1.2, CI 95%). The use of rivaroxaban is associated with the highest risk of GIT bleeding, while the use of apixaban appears to be the safest [16,17,22].

All of the above-mentioned meta-analyses and studies were retrospective and evaluated only major and clinically significant GIT bleeding events. Our work was prospective with the exclusion of patients with pre-existing lesions on the upper GIT mucosa, which could increase the possibility of subsequent bleeding after initiating anticoagulant therapy. If anticoagulant therapy was initiated with intact upper gastrointestinal mucosa, no serious or clinically significant bleeding was observed in any of our patients (NOACs or warfarin). None of our monitored patients needed hospitalization or non-scheduled medical examination within three months after the initiation of therapy. We have not confirmed a more frequent occurrence of bleeding with dabigatran treatment, although patients from this group had a higher average HAS-BLED score than the other patient groups. However, with rivaroxaban, the bleeding rate may have been decreased by concomitant PPI therapy. The difference in the incidence of bleeding overall as well as the occurrence of all endoscopically observed mucosal bleeding events in the upper GIT after three months of treatment was statistically insignificant when comparing individual drugs, although it was quite high (in 20% of patients treated with warfarin, in 20% on dabigatran, in 10% on rivaroxaban, and in 5% treated with apixaban). Therefore, it seems appropriate to carry out EGD examination in all risk patients before starting anticoagulant therapy, particularly in patients receiving NSAIDs or with a history of gastroduodenal ulcer disease or upper GIT dyspeptic symptoms. 

When comparing the incidence of bleeding with concomitant treatment with PPI, the difference among groups was non-significant, although we found a trend towards a reduction in bleeding in patients receiving PPI. Similarly, a higher incidence of upper GIT bleeding in patients with a history of gastroduodenal ulcer disease not receiving PPI was consistent with the data of Maruyama et al. [3]. Chan et al. [23] reported a 50% reduction of the bleeding risk in patients treated with PPI and dabigatran. There is an ongoing discussion regarding interactions between PPI and dabigatran that may lead to a decrease in the dabigatran plasma level and thus a reduction in gastric bleeding [24]. Thus, PPI appears as a key medication in preventing upper GIT bleeding during NOAC treatment, especially in patients at an increased risk of GIT bleeding [13]. In our patients, PPI was preventively administered in 19 of 26 patients with a history of gastroduodenal ulcer disease or GERD, and only two of them (10.5%) had mucosal clinically asymptomatic bleeding. 

Age over 75 years is generally reported as a risk factor for increased GIT bleeding [2,8,22,25], but this trend was not observed in our series. We also did not record an increased incidence of bleeding depending on the dose of NOACs. This observation may be explained by the fact that the dose of NOACs has been reduced in patients over 75 years of age according to the protocol. 

The presence of HP was shown as a risk factor for upper GIT bleeding (Spearman’s correlation coefficient *R* = 0.528, *p* < 0.05). Despite the low incidence of HP (only 14.5%), endoscopic signs of bleeding were observed in two thirds of these patients. In four patients, we found erosive changes in the upper GIT, which were always associated with bleedings signs. Therefore, it seems appropriate to detect and eradicate HP in patients treated with NOAC or warfarin in order to reduce the possibility of erosion and ulceration in the upper GIT and consequent bleeding.

According to current ESGE recommendations [26], biopsies in the GIT during warfarin treatment are safe provided the INR lies within the therapeutic range of 2–3. In patients on NOACs, it is recommended to withdraw therapy 24 h prior to endoscopic examination and biopsy. We performed biopsies to detect HP in 12 patients treated with warfarin and 41 patients treated with uninterrupted dosage of NOACs. No post-biopsy bleeding was observed in either group of patients, allowing us to conclude that biopsies are safe in the context of an NOAC treatment event without the discontinuation of drugs.

## 5. Limitations

This study has certain limitations. The number of patients in each group was relatively small, and the follow-up period was short, only 3 months. Despite randomization, there was a difference in the HAS-BLED score between the warfarin and dabigatran group, which could skew the incidence of bleeding after the treatment period in favor of warfarin. Still, we do not expect that these limitations would fundamentally distort the results of the study.

## 6. Conclusions

We prospectively investigated the effects of NOAC on the upper gastrointestinal mucosa in patients treated with dabigatran, rivaroxaban, and apixaban and compared them with the effects of warfarin. After the exclusion of pre-existing organic lesions at the initial EGD examination, within three months after the initiation of anticoagulant treatment, we did not find any difference in the bleeding incidence between NOACs themselves or NOACs and warfarin.

The risk factor for upper GIT bleeding was the presence of HP, while PPI administration had a protective effect. The incidence of bleeding in our group of patients was not affected by age, NOAC dose, or the presence of DM.

Therefore, EGD examination prior to anticoagulant therapy also targeted at evaluating the presence of HP as well as preventive administration of PPI seems appropriate in patients with increased risk of upper GIT bleeding.

We did not see any continued bleeding associated with biopsy without the discontinuation of anticoagulant therapy, including NOACs, supporting the safety of this method.

## Figures and Tables

**Figure 1 medicina-56-00363-f001:**
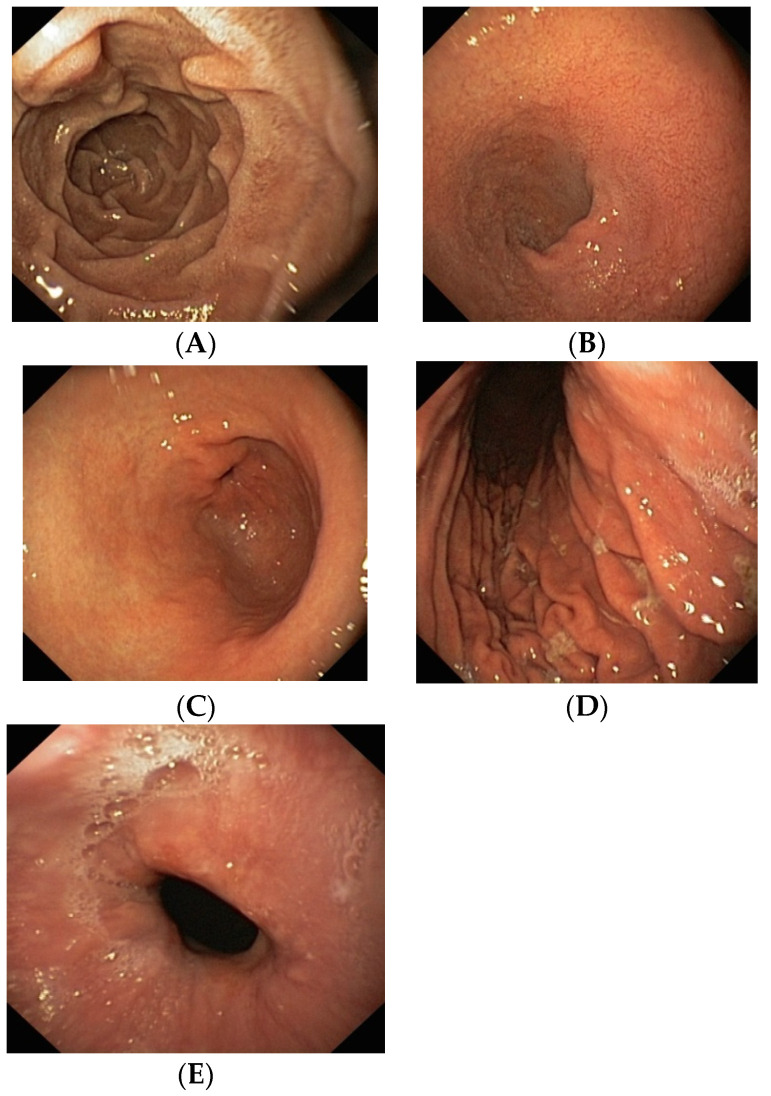
Normal endoscopic findings in patients before anticoagulant therapy. (**A**)—D2 duodenum, (**B**)—duodenal bulbus, (**C**)—stomach antrum, (**D**)—stomach body, (**E**)—distal esophagus.

**Figure 2 medicina-56-00363-f002:**
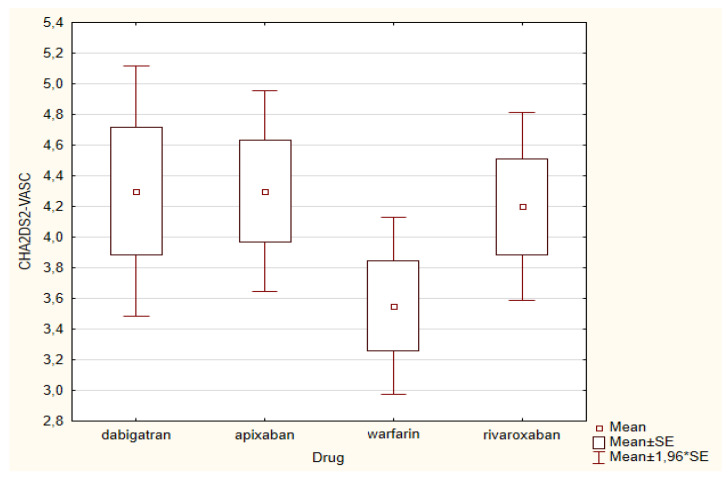
Comparison of the CHA_2_DS_2_-VASc score in groups of patients according to the individual drugs. Depicted are means and standard errors of the mean. No significant difference among groups was found.

**Figure 3 medicina-56-00363-f003:**
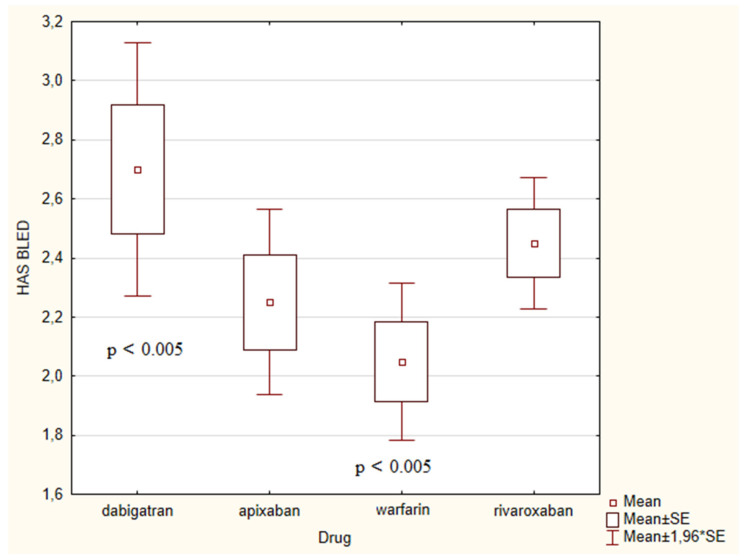
Comparison of the HAS-BLED score in groups of patients according to the individual drugs. Depicted are means and standard errors of the mean. A group of patients on warfarin had a lower HAS-BLED score com*pared to the dabigatran group.

**Figure 4 medicina-56-00363-f004:**
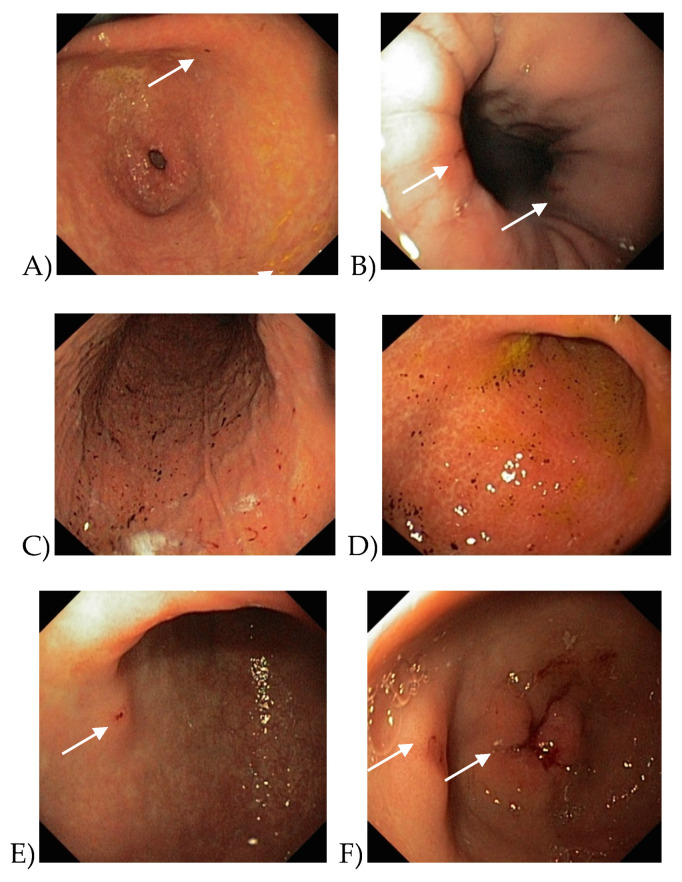
Endoscopic findings in patients with bleeding. (**A**)—sporadic petechiae in the stomach antrum during dabigatran treatment. (**B**)—Cameron’s lesions with bleeding signs during dabigatran treatment. (**C**,**D**)—examples of hemorrhagic gastropathy during warfarin treatment. (**E**)—erosions in the stomach antrum with bleeding signs during rivaroxaban treatment. (**F**)—erosions in the stomach antrum with bleeding signs during apixaban treatment. Arrows show the sites of bleeding.

**Table 1 medicina-56-00363-t001:** Patient characteristics and medical history.

	Age	Sex	UV	HP	NSAID-SA	PPI	DM	MI	NYHA	AH	TIA, Stroke, PAE	PAO	CHA_2_DS_2_-VASC	HAS-BLED	Indication	Bleeding into GIT/all	Biopsy
Warfarin	72.4 (60–86)	F 9/M 11	6	2/15	0	GERD 5/UV 2	3	3	2.1 ± 0.31	20	4	3	3.55 ± 1.32	2.05 ± 0.60	DVT 4 /FA 16	4 GIT/6 all	12
Dabigatran	73.15 (60–89)	F 7/M 13	7	2/15	0	GERD 2/UV 3	6	3	2.2 ± 0.41	20	7	1	4.30 ± 1.87	2.7 ± 0.98	DVT 4 /FA 16	4 GIT/4 all	13
Rixaroxaban	77.25 (67–93)	F 15/M 5	10	2/17	2	GERD 8/UV 6	2	4	2.15 ± 0.37	20	5	0	4.20 ± 1.39	2.45 ± 0.51	DVT 4 /FA 16	2 GIT/4 all	12
Apixaban	76.35 (66–88)	F 11/M9	3	3/15	0	GERD 6/UV 3	10	3	2.1 ± 0.31	20	5	4	4.30 ± 1.49	2.25 ± 0.72	DVT 0 /20 FA	1 GIT/3 all	16

Abbreviations: F—female, M—male, HP—*Helicobacter pylori*, NSAID—nonsteroidal anti-inflammatory drug, ASA—acetylsalicylic acid, PPI—proton pump inhibitors, GERD—gastroesophageal reflux disease, UV—ulcus ventriculi, DM—diabetes mellitus, MI—myocardial infarction, NYHA—class of heart failure, AH—arterial hypertension, TIA—transient ischemic attack, PA—pulmonary artery, PAO—peripheral arterial disease, DVT—deep vein thrombosis, AF—atrial fibrillation, GIT—gastrointestinal tract.

**Table 2 medicina-56-00363-t002:** Overview of sources of bleeding after 3 months of anticoagulant treatment according to specific groups.

	Warfarin	Dabigatran	Rivaroxaban	Apixaban
Upper gastrointestinal Tract	Petechiae in stomach antrum, HP negative	Petechiae in stomach antrum, HP positive	Erosions in stomach antrum with bleeding signs, HP positive	Ulcus ventriculi, petechiae in stomach antrum, HP positive
	Petechiae in stomach antrum, HP negative	Petechiae in stomach antrum, HP negative	Erosions in stomach antrum with bleeding signs, HP positive	
	Haemorrhagic gastropathy, HP positive	Cameron’s lesions with bleeding signs in hiatus hernia		
	Haemorrhagic gastropathy, HP negative	Erosions in stomach antrum with bleeding signs, HP positive		
Location other than the upper gastrointestinal tract	Positive FOBT–EGD and colonoscopy negative		Epistaxis	Enterorrhagia from haemorrhoids
	Positive FOBT–EGD and colonoscopy negative		Positive FOBT–EGD and colonoscopy negative	Ppositive FOBT–EGD and colonoscopy negative

Abbreviations: FOBT—fecal occult blood test. HP—*Helicobacter pylori*, EGD—endogastroduodenoscopy.

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
