# Peer review of "The Impact of Novel Anticoagulants on the Upper Gastrointestinal Tract Mucosa"

_medicina, 2020, doi:10.3390/medicina56070363_

Round 1

Reviewer 1 Report

The authors performed an interesting paper on the impact of novel anticoagulants on the upper gastrointestinal tract mucosa compared to warfarin

Some important information are missing from the text.

Are the authors gastroenterologists?

Where they work?

Which is the hospital where the study was done?

Who provided the patients, hematologist cardiologist?

Who did the endoscopic exam? Always the same operator?

Were endoscopist experienced operators?

Specify the characteristics of the endoscope

It would be interesting to put a diagram of the patient enrollment algorithm and the drop outs

The DOAC dosages must be specified

Tables and figures should be placed at the end of the work so as not to stop reading the text

Add the ethics committee approval number. Has it been registered to trial gov?

In table 1 it would be better to put only the characteristics of the patients at enrollment and in table two the endoscopic results after three months.

Why didn't you exclude patients on asa therapy or NSAID? , this can be a confounding factor

On line 109 what does it mean that the first 20 pts with a therapeutic value of INR were taken into evaluation do you mean that you have had drop outs or excluded patients?

Table 1: What does it mean that 6 patients in warfarin group have ulcer dis? And 10 pts in rivaroxaban? do the authors mean that they had a history of ulcer in the past but at the time of enrollment had a normal endoscopic examination?

In table 1: what does the biopsy column refer to? Why in only 13 patients of rivaroxaban, in 12 of dabigatran and 16 of apixaban biopsies were done?

In Figures 2-3- the commercial names of the DOACs must be replaced with the active principle, as in the text

In table 2 the authors put the results of FOBT but in the text it was never mentioned the fact that some patients or all of them did the search for occulted blood in the stool

The conclusions (line 222) are not supported by the results considering that the NOAC dosages have not been specified in the text

Have you followed up these patients over time? For example with hemoglobin level? How could you conclude that there was no bleeding after the biopsy?

Please add the limitation of the study

Author Response

Please see the attachcment.

Reviewer 2 Report

        Gastrointestinal bleeding is increasingly common, considering the increasing use of oral anticoagulants for prevention of pulmonary embolism due to venous thrombosis or systemic embolism due to atrial fibrillation. Metanalyses have suggested that NOACs may have an increased gastrointestinal bleeding risk compared to warfarin, which is thought to be driven by Rivaroxaban and Dabigatran. In the present study, the authors evaluated the impact of oral anticoagulant therapy on the upper gastrointestinal mucosa. They included a total of 80 patients, without pre-existing gastroesophageal lesions, 20 patients in each oral anticoagulant group (warfarin, dabigatran, rivaroxaban and apixaban). Within three months after the initiation of anticoagulant treatment, the authors did not find any difference in bleeding incidence between these four anticoagulants. Also, the PPI administration had a protective effect while the presence of HP is a risk factor for upper gastrointestinal bleeding.

Main comments:

  1. The authors must define very clearly the inclusion criteria of the patients and also the clinic from which they chose the patients. They mentioned that they included 20 patients in each group, without specifying how they were chosen (were there exactly 20 patients in each group without pre-existing gastroesophageal lesions or some patients were excluded?).
  2. I consider that the authors should mention how many of the patients in each group were receiving antiplatelet therapy and what it was.
  3. In lines 123-128, the authors mentioned that the highest HAS-BLED score was found in patients treated with dabigatran and the lowest in those treated with warfarin and that no difference was found between CHADS-VASc score among groups. This should be discussed considering that this may influence the final results of the study. Also, the authors should discuss their results by comparing with those from a recent systematic review of real-world, non-industry sponsored studies published by Anghel et al. (,,Non-vitamin K Antagonist Oral Anticoagulants and the Gastrointestinal Bleeding Risk in Real-World Studies”, published in Journal of Clinical Medicine, May 2020, doi: 3390/jcm9051398). They found that the lowest risk of GIB was with apixaban when compared to warfarin and apixaban was also associated with a lower risk of GIB compared to dabigatran or rivaroxaban.
  4. The authors concluded that PPI administration had a protective effect on GIB. The authors consider that all patients with oral anticoagulant treatment should also receive a PPI, regardless of the presence or absence of pre-existing gastroesophageal lesions?
  5. From line 172 to line 181, the authors should remove the names of the authors from the text and leave only the number of the reference.

Round 2

Reviewer 1 Report

The authors have added what is required in the text